DISCOVERY REPORT

# A lifecourse mendelian randomization study highlights the long-term influence of childhood body size on later life heart structure

**Katie O'Nunain**[1], **Chloe Park**[2], **Helena Urquijo**[1,3], **Genevieve M. Leyden**[1,3,4], **Alun D. Hughes**[2], **George Davey Smith**[1,3], **Tom G. Richardson**[1,3,5]*

**1** Bristol Medical School, University of Bristol, Oakfield House, Oakfield Grove, Bristol, United Kingdom, **2** MRC Unit for Lifelong Health and Ageing at UCL, Department of Population Science and Experimental Medicine, Institute of Cardiovascular Science, University College London, London, United Kingdom, **3** MRC Integrative Epidemiology Unit (IEU), Population Health Sciences, Bristol Medical School, University of Bristol, Oakfield House, Oakfield Grove, Bristol, United Kingdom, **4** Bristol Medical School: Translational Health Sciences, Dorothy Hodgkin Building, University of Bristol, Bristol, United Kingdom, **5** Novo Nordisk Research Centre, Headington, Oxford, United Kingdom

* Tom.G.Richardson@bristol.ac.uk

The Editors encourage authors to publish research updates to this article type. Please follow the link in the citation below to view any related articles.

## Abstract

Children with obesity typically have larger left ventricular heart dimensions during adulthood. However, whether this is due to a persistent effect of adiposity extending into adulthood is challenging to disentangle due to confounding factors throughout the lifecourse. We conducted a multivariable mendelian randomization (MR) study to separate the independent effects of childhood and adult body size on 4 magnetic resonance imaging (MRI) measures of heart structure and function in the UK Biobank (UKB) study. Strong evidence of a genetically predicted effect of childhood body size on all measures of adulthood heart structure was identified, which remained robust upon accounting for adult body size using a multivariable MR framework (e.g., left ventricular end-diastolic volume (LVEDV), Beta = 0.33, 95% confidence interval (CI) = 0.23 to 0.43, $P = 4.6 \times 10^{-10}$). Sensitivity analyses did not suggest that other lifecourse measures of body composition were responsible for these effects. Conversely, evidence of a genetically predicted effect of childhood body size on various other MRI-based measures, such as fat percentage in the liver (Beta = 0.14, 95% CI = 0.05 to 0.23, $P = 0.002$) and pancreas (Beta = 0.21, 95% CI = 0.10 to 0.33, $P = 3.9 \times 10^{-4}$), attenuated upon accounting for adult body size. Our findings suggest that childhood body size has a long-term (and potentially immutable) influence on heart structure in later life. In contrast, effects of childhood body size on other measures of adulthood organ size and fat percentage evaluated in this study are likely explained by the long-term consequence of remaining overweight throughout the lifecourse.

## Introduction

The prevalence of childhood obesity has increased rapidly in the last 50 years, and it is now a major public health concern worldwide [1]. Research suggests that childhood obesity has serious

**Data Availability Statement:** Summary statistics from genome-wide association studies can be found as referenced in S1 Table and additional underlying data supporting this paper can be found within S2–S16 Tables. All individual level data analysed in this study can be accessed via an approved application to ALSPAC (http://www.bristol.ac.uk/alspac/researchers/access/) and the UK Biobank study (https://www.ukbiobank.ac.uk/enable-your-research/apply-for-access).

**Funding:** This work was supported by the Integrative Epidemiology Unit which receives funding from the UK Medical Research Council and the University of Bristol (MC_UU_00011/1). AH received support from the Wellcome Trust (086676/7/08/Z) and the British Heart Foundation (PG/06/145, CS/15/6/31468 & SP/F/21/150020). The funders had no role in study design, data collection and analysis, decision to publish, or preparation of the manuscript.

**Competing interests:** I have read the journal's policy and the authors of this manuscript have the following competing interests: TGR is employed part-time by Novo Nordisk outside of this work. All other authors declare no conflicts of interest.

**Abbreviations:** ALSPAC, Avon Longitudinal Study of Parents and Children; BMI, body mass index; CI, confidence interval; FFMI, fat-free mass index; GRACE, Growth Related effects in ALSPAC on Cardiac Endpoints; GWAS, genome-wide association studies; IVW, inverse variance weighted; LVEF, left ventricular ejection fraction; LVEDV, left ventricular end-diastolic volume; LVESV, left ventricular end-systolic volume; MR, mendelian randomization; MRI, magnetic resonance imaging; OR, odds ratio; SAT, subcutaneous adipose tissue; SV, stroke volume; UKB, UK Biobank; VAT, visceral adiposity tissue.

long-term health consequences including increased risk of cardiovascular disease in adulthood [2–4]. This has prompted efforts into understanding the effects of childhood obesity on cardiac structure and function in later life, with previous studies noting an association between childhood adiposity and both left ventricular remodeling and left ventricular mass in adulthood [5–9].

However, evidence of an association between childhood obesity and altered cardiac morphology comes from observational studies, which are prone to confounding and reverse causation. This is the motivation behind an approach known as mendelian randomization (MR), a form of instrumental variable analysis that harnesses genetic variants randomly allocated at birth to investigate evidence of a causal effect between modifiable lifestyle risk factors on complex traits and disease outcomes [10,11]. Therefore, as long as the assumptions of MR hold, differences in an outcome between carriers of specific genetic variants and noncarriers can be attributed to the environmental risk factors they predict.

Multivariable MR is an extension of the conventional MR approach that simultaneously estimates the genetically predicted effects of multiple risk factors on an outcome [12,13]. This approach can help separate the "direct" and "indirect" effects of a risk factor on an outcome (**Fig 1**). Recently, we derived sets of genetic variants to separate the genetically predicted effects of childhood and adult body size using multivariable MR in a lifecourse context [14]. These scores have already been validated to separate measured childhood and adult body mass index (BMI) [15,16] and have also been leveraged to provide evidence that childhood body size has a direct influence on outcomes such as type 1 diabetes [17] (**Fig 1B**). In contrast, these scores have provided evidence of an indirect effect on outcomes such as atherosclerosis and heart failure [18] (**Fig 1C**), suggesting that the association between childhood adiposity and these outcomes is likely attributed to individuals remaining overweight into adulthood. However, this approach has not yet been applied to evaluate the effect of childhood body size on cardiac structure and function in later life, which is vital in terms of understanding the long-term consequences of this early life exposure on the cardiovascular system.

In this study, we applied univariable and multivariable MR to investigate whether genetically predicted childhood body size has a direct effect on magnetic resonance imaging (MRI) assessed measures of cardiac structure and function in adulthood independent of adult body size. Although genetic instruments for childhood body size were derived as a surrogate measure of adiposity, we investigated this using various sensitivity analyses to evaluate whether they could be explained by other lifecourse measures of body composition. We next applied univariable and multivariable MR to other MRI-derived measures of abdominal organs measured during adulthood, involving the size and fat percentage of the liver, pancreas, and kidney, as well as volumes of subcutaneous adipose tissue (SAT) and visceral adipose tissue (VAT). These abdominal traits were analyzed for comparative purposes, given that we anticipated there to be weak evidence of an effect of childhood body size on them upon accounting for the effect of adult body size. Last, we analyzed cardiomyopathy endpoints using this approach to discern whether putative effects responsible for left ventricular cardiac remodeling may have downstream implications on this disease outcome.

## Results

### Investigating the direct and indirect effects of childhood body size on cardiac structure and function in later life

An overview of the datasets analyzed in this study and their study characteristics can be found in **S1 and S2 Tables**, respectively. Univariable MR analyses using the inverse variance weighted (IVW) approach provided strong evidence that childhood body size has a total effect on left ventricular end-diastolic volume (LVEDV) (Beta = 0.36 SD change per change in body

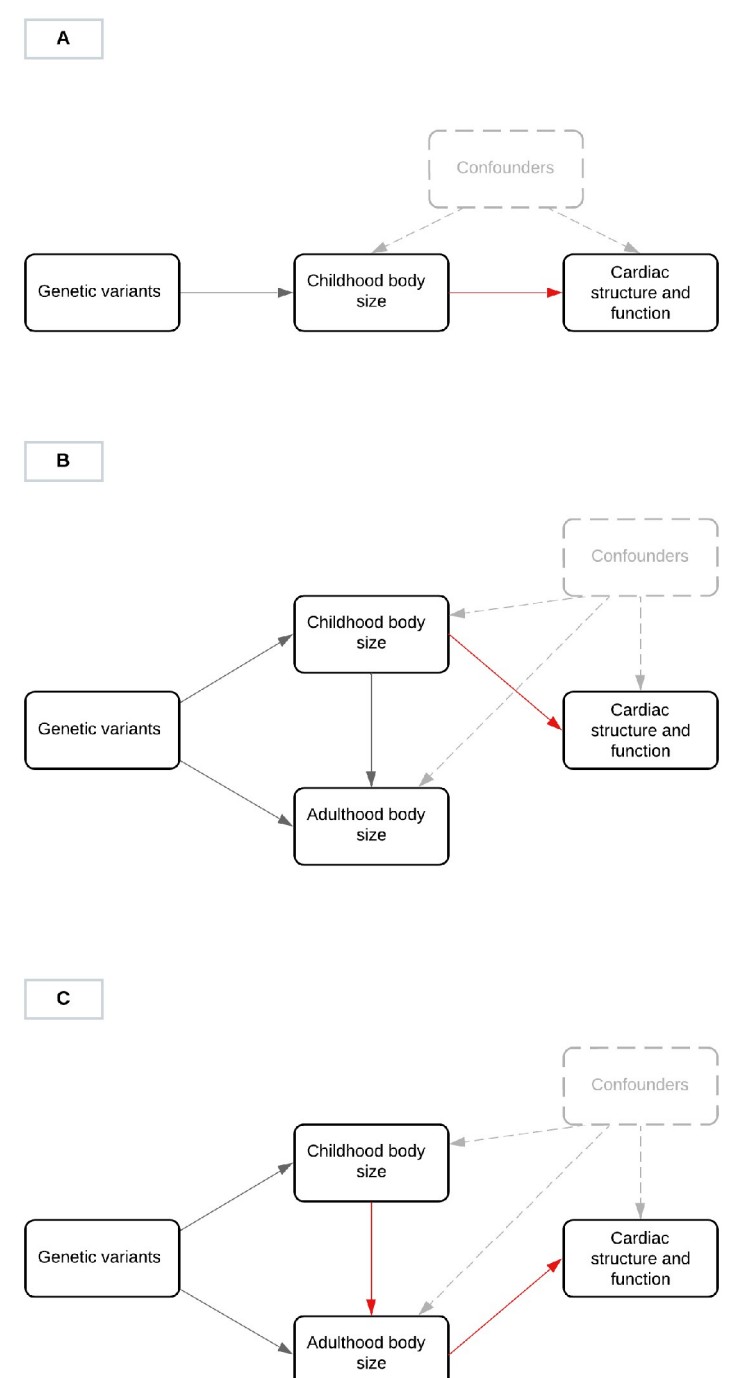

**Fig 1. DAGs illustrating the different scenarios through which childhood body size may influence cardiac structure in later life. Fig 1A** illustrates the "total" effect of childhood body size on cardiac structure in adulthood. This may be due to a "direct" effect of childhood body size, which is depicted in **Fig 1B** or an "indirect" effect, mediated through adult body size, which is depicted in **Fig 1C**. DAG, directed acyclic graph.

size category, 95% confidence interval (CI) = 0.28 to 0.44, $P = 1 \times 10^{-18}$), left ventricular end-systolic volume (LVESV) (Beta = 0.29, 95% CI = 0.21 to 0.36, $P = 3 \times 10^{-13}$), and stroke volume (SV) (Beta = 0.36, 95% CI = 0.28 to 0.45, $P = 1 \times 10^{-16}$). However, there was weak evidence of an effect on left ventricular ejection fraction (LVEF) (Beta = −0.10, 95% CI = −0.18 to −0.02, $P = 0.016$) after accounting for multiple testing corrections across all MRI-based measures in this study ($P < 0.0045$). Similar results were observed for adult body size in the univariable analysis, although effect estimates were typically smaller in magnitude (S3 Table). In addition, childhood body size estimates on measures of heart structure were supported by the weighted median and MR–Egger methods suggesting that our results were robust to horizontal pleiotropy, whereas weak evidence was identified on LVEF when using these approaches (S3 Table).

Multivariable MR analyses provided strong evidence of a direct effect of childhood body size on cardiac measures in adulthood (S4 Table) as effect estimates for LVESV (Beta = 0.29, 95% CI = 0.19 to 0.40, $P = 8 \times 10^{-8}$), LVEDV (Beta = 0.33, 95% CI = 0.23 to 0.43, $P = 5 \times 10^{-10}$), and SV (Beta = 0.31, 95% CI = 0.20 to 0.41, $P = 1 \times 10^{-8}$) remained robust upon accounting for adult body size. Furthermore, multivariable estimates provided little evidence for a direct effect of adult body size independent of childhood body size on cardiac structure when analyzed in the multivariable framework along with childhood body size (S4 Table). Forest plots for both the univariable and multivariable MR results on measures of cardiac structure and function can be found in Fig 2.

Validation analyses conducted in the ALSPAC cohort supported a direct effect of childhood body size on measures of cardiac structure at mean age 17.8 years in the lifecourse (S5 Table). In the multivariable MR analyses, childhood body size provided strong evidence of an effect on LVEDV (Beta = 1.65ml per change in body size category, 95% CI = 0.50 to 2.80, $P = 0.005$), LVESV (Beta = 0.75ml, 95% CI = 0.11 to 1.38, $P = 0.022$) and SV (Beta = 0.89ml, 95% CI = 0.19 to 1.59, $P = 0.013$). Weak evidence of an effect of childhood body size was found when analysing LVEF (Beta = −0.07, 95% CI = −0.42 to 0.29, $P = 0.715$) as found in our primary analysis.

## Evaluating the direct and indirect effects between childhood body size and abdominal organ size in adulthood

We then applied the same approach to MRI measures of abdominal organs in adulthood for comparison. Univariable MR provided evidence of an effect of child and adult body size on all measures of abdominal organ size and fat percentage with the exception of pancreatic volume (S6 Table). For example, there was strong evidence of a total effect of childhood body size on kidney volume (Beta = 0.36, 95% CI = 0.27 to 0.46, $P = 1 \times 10^{-13}$), liver volume (Beta = 0.41, 95% CI = 0.32 to 0.51, $P = 5 \times 10^{-17}$), pancreatic fat percentage (Beta = 0.21, 95% CI = 0.10 to 0.33, $P = 3 \times 10^{-4}$), and liver fat percentage (Beta = 0.14, 95% CI = 0.05 to 0.23, $P = 0.002$) using the IVW method. However, the evidence of an effect for child body size drastically attenuated in the multivariable MR analysis accounting for adult body size (with the direction of effect for childhood body size even reversing in some instances). This suggests that child body size acts indirectly through adult body size on abdominal organ size and fat percentage in later life (S7 Table). In addition, there was also strong evidence of a direct effect of adult body size on SAT (Beta = 0.97, 95% CI = 0.87 to 1.07, $P = 6 \times 10^{-84}$) and VAT volume (Beta = 0.80, 95% CI = 0.71 to 0.90, $P = 1 \times 10^{-62}$). All univariable and multivariable MR estimates on abdominal traits are shown in Fig 2.

## Incorporating the genetically predicted effects of other measures of lifecourse body composition on cardiac structure

Repeating multivariable MR analyses for childhood body size while accounting for adult fat-free mass index (FFMI) in the model continued to provide evidence of a direct effect of

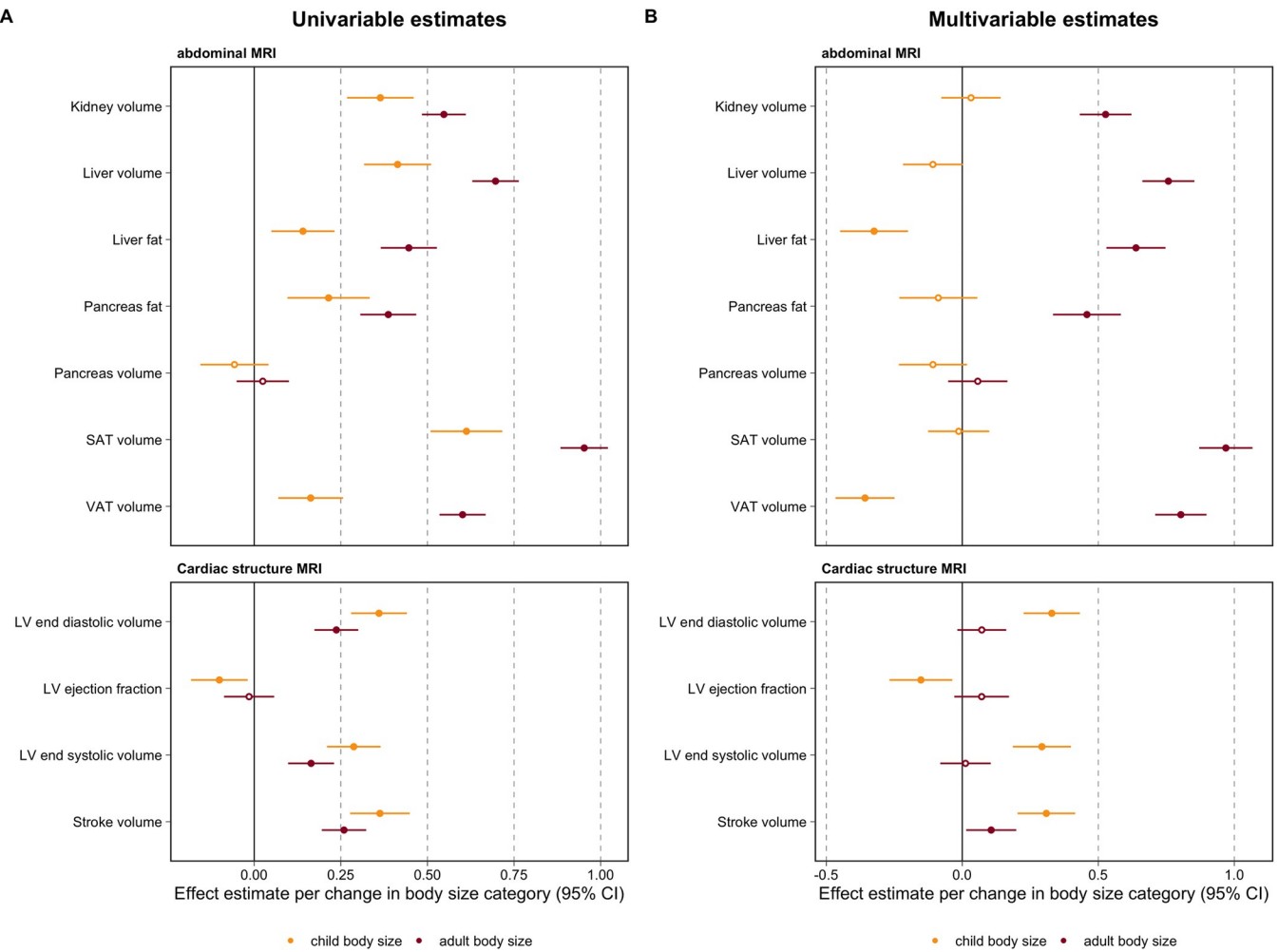

**Fig 2. Forest plots illustrating (A) univariable and (B) multivariable MR effect estimates of childhood and adult body size on measures of cardiac structure/function and abdominal organ size/fat percentage.** The estimates for child body size are in orange and the estimates for adult body size are in red. The effect estimates are per change in body size category and include the 95% CI. The data underlying this figure can be found in S3, S4, S6, and S7 Tables. CI, confidence interval; LV, left ventricular; MR, mendelian randomization; MRI, magnetic resonance imaging; SAT, subcutaneous adipose tissue; VAT, visceral adiposity tissue.

childhood body size on cardiac structure and function in later life (S8 and S9 Tables). Likewise, strong evidence of a genetically predicted effect of childhood body size on measures of cardiac structure was found upon accounting for birth weight using multivariable MR (S10 and S11 Tables). Forest plots for the univariable and multivariable MR analyses accounting for FFMI and birth weight can be found in S1 and S2 Figs, respectively.

Repeating MR analyses using childhood and adult height as our exposures provided strong evidence on all 4 MRI-assessed measures of cardiac structure and function (S12 Table). For example, we observed evidence of an effect of height using the IVW method at both the childhood (Beta per change in height category = −0.26, 95% CI = −0.31 to −0.21, $P = 1 \times 10^{-25}$) and adult time points (Beta = −0.31, 95% CI = −0.36 to −0.26, $P = 1 \times 10^{-34}$) on LVEF. However, in contrast to our findings for childhood body size, multivariable MR found that evidence of an effect for childhood height on LVEF attenuated drastically and upon accounting for adult height (Beta = −0.05, 95% CI = −0.19 to 0.10, $P = 0.53$). This suggests that childhood height exerts its effect on LVEF indirectly via the causal pathway involving adult height, but also that

our findings for childhood body size may be more likely to be due to higher adiposity as opposed to simply being larger during childhood. Evidence on measures of cardiac structure also typically attenuated for childhood height in comparison to adult height (**S13 Table**). Forests plots of the MR results for childhood and adult height are depicted in **S3 Fig.** Finally, evidence of an effect of childhood body size on measures of cardiac structure remained strong in the multivariable model accounting for the effect of childhood height (**S14 Table**).

## Weak evidence that childhood body size directly influences risk of cardiomyopathies in adulthood

Despite strong evidence of an effect on cardiac structure provided by previous analyses, undertaking the same analytical approach on cardiomyopathy endpoints provided weak evidence that childhood body size has a direct effect on these disease outcomes (**S15** and **S16** **Tables**). For example, the total effect of childhood body size found in univariable MR analyses of nonischemic cardiomyopathy (odds ratio (OR) = 1.74 per change in body size category, 95% CI = 1.20 to 2.53, $P = 0.004$) attenuated to include the null in the multivariable MR analyses accounting for adult body size (OR = 1.09, 95% CI = 0.64 to 1.84, $P = 0.753$).

## Discussion

In this study, we provide evidence that childhood body size directly influences cardiac structure in later life independent of adult body size. Furthermore, our effect estimates remained robust even after accounting for genetically predicted lean mass and birth weight, further supporting the hypothesis that childhood body size has an independent effect on cardiac structure. In contrast, there was weak evidence of an effect of childhood body size on LVEF, consistent with findings from the literature suggesting that obesity may influence cardiac remodeling [19]. Additionally, as anticipated evidence of a genetically predicted effect of childhood body size on adult measures of abdominal organ size and fat percentage attenuated after accounting for body size during adulthood. These results suggest that the total effect of childhood body size is likely attributed to the long-term consequence of remaining overweight throughout the lifecourse and into adulthood. Likewise, although childhood body size increased the risk of cardiomyopathy there was no convincing evidence that this is due to a direct effect (i.e., independently of adult body size).

Previous studies have used cardiac MRI to investigate the effect of childhood adiposity on cardiac structure and function during childhood [20,21]. They report evidence of an association between childhood adiposity and increased left ventricular mass and cardiac remodeling. Findings from our study provides evidence using genetic instrumental variables that these reported associations may be due to a direct effect of childhood body size on cardiac structure. One potential mechanism that has been postulated for this finding is higher levels of adipose tissue in early life increasing circulating blood volume and cardiac output [22,23]. These hemodynamic changes in combination with other metabolic and neurohormonal alterations are thought to drive changes in cardiac morphology [24,25]. Another proposed mechanism that this finding may be attributed is increased early life body size resulting in a persistent change in myocardial energetics [26]. Cardiac remodeling can be a normal physiological process; however, it has also been reported to potentially become irreversible [27,28]. We note, however, that, although our findings highlight the importance of body size during early life as a determinant of cardiac structure in adulthood, further research is required to pinpoint the critical windows during the lifecourse when the consequence of this effect may become immutable.

The genetically predicted effects of childhood body size on cardiac remodeling observed in our study also required further investigation into whether they may lead to pathological consequences and if this translates into an increased risk of cardiovascular disease. The current literature suggests that childhood adiposity influences cardiometabolic disease risk only if the levels remain consistently high into adulthood [29]. Of particular note is a recent MR study which found that the effect estimates for childhood body size and 8 cardiovascular disease endpoints attenuated (and in some cases even reversed direction of effect) when accounting for adult body size [18]. We also build on these findings in this study, as evidence that childhood body size increases risk of nonischaemic cardiomyopathy from our univariable analyses did not remain robust to the inclusion of adult body size in the multivariable model. These findings suggest that individuals who are larger in early life are likely at higher risk of nonischaemic cardiomyopathy in later life due to a sustained and long-term effect of adiposity for many years across the lifecourse. However, this research question would be worthwhile revisiting once larger number of cases for cardiomyopathy endpoints are available [30]. Moreover, investigations into the consequences of body size at other time points in the lifecourse would be worthwhile, particularly given that previous observational analyses suggest that adiposity in late adolescence (mean age 18.3 years) may contribute to being diagnosed with cardiomyopathy in adulthood [31]. We also assessed effects on left ventricular function in this study using left ventricular ejection fraction, although alternate measures may be worthwhile investigating once larger sample sizes are available [32].

We additionally incorporated genetic instruments for FFMI, birth weight, and height (during both childhood and adulthood) into our multivariable MR framework to investigate whether these might explain the genetically predicted effect found between childhood body size and measures of cardiac structure. Although previous studies have indicated that cardiac structure is more strongly influenced by lean mass than fat mass, our effect estimates remained robust when accounting for FFMI in adulthood [33]. In addition, the effect estimates for LVESV, LVEDV, and SV did not attenuate when birth weight was incorporated into the multivariable model. These results support the hypothesis that childhood body size has an effect on adult cardiac structure independent of birth weight used as a proxy in this study for body size during the very early stages in the lifecourse. However, future research that incorporates both parental and fetal genotypes into the study design would be more appropriate to fully evaluate the genetically predicted effect of birth weight itself on MRI-derived traits such as cardiac structure [34,35]. Furthermore, the genetically predicted effect of childhood height on cardiac structure did not remain robust after accounting for height during adulthood. Taken together, the evidence of a genetically predicted effect of childhood body size on cardiac structure found in this study may be driven by adiposity rather than these alternate aspects of body composition, although confirmatory evidence from further research is required to support this.

It is important to note that this study has limitations. First, to gain a large number of reliable instrumental variables for childhood body size, we harnessed recall data [36]. However, as mentioned in the methods section, these genetic variants have been validated in 3 separate studies and have even been found to be a better predictor of BMI across multiple time points in childhood compared to the genetic score from derived from the largest genome-wide association study (GWAS) of measured childhood BMI to date [37]. Furthermore, although the UK Biobank is by far the largest study to date with MRI measures of cardiac structure and function, the subsample of participants who attended the MRI imaging study have been reported to have a "healthy bias" [38], and these individuals were removed from our GWAS analyses required for instrument derivation. However, this was necessary to prevent overlap between our exposures and outcomes that may induce overfitting into MR analyses and lead estimates away from the null [39].

In conclusion, our findings suggest that childhood body size has a direct and potentially immutable effect on cardiac structure in later life. This is in contrast to results for abdominal organ size and fat percentage, where associations with childhood obesity are likely explained by a persistent effect of adiposity throughout the lifecourse into adulthood. Further research is needed to determine whether early life changes in cardiac morphology caused by childhood body size have pathological consequences.

## Materials and methods

### Data resources

**Genetic instruments for childhood and adult body size.**    We previously conducted GWASs in the UKB study on measures of childhood and adult body size. Details of these analyses have been reported elsewhere [14]. In brief, the childhood body size measure in UKB was derived using recall questionnaire data asking participants if they were "thinner," "plumper," or "about average" when they were aged 10 years old compared to the average (field #1687). Adult measured BMI (field #21001) data (mean age 56.5 years) was then transformed into a 3-tier variable using the same proportions as the childhood measure for comparative purposes. Genetic instruments derived from these GWASs have been previously validated using measured BMI data from 3 independent populations; the Avon Longitudinal Study of Parents and Children (ALSPAC) [14], the Trøndelag Health (HUNT) study [15], and the Cardiovascular Risk in Young Finns Study [16]. Furthermore, genetic correlation analyses demonstrate that the childhood body size GWAS is much more highly correlated with measured childhood obesity from an independent sample (rG = 0.85) compared to the adult measure (rG = 0.67). In contrast, results from the adult body size GWAS have been shown to be much more strongly correlated with measured BMI in adulthood (rG = 0.96) compared to the childhood measure (rG = 0.64). Conditional F-statistics generated for childhood (F = 13.6) and adult (F = 16.0) body size instruments suggested that weak instrument bias was unlikely for these sets of genetic variants.

In the current study, we repeated these GWASs in UKB excluding participants who attended UKB assessment centers for MRI data collection. As these MRI measures were analyzed as outcomes in this study, this allowed us to partition UKB into 2 independent samples, meaning there was no sample overlap between our exposures and outcomes which may lead to overfitting in MR analyses [39,40]. GWASs were conducted on $n$ = 407,741 participants with both measures adjusting for age, sex, and genotyping chip, with the childhood body size GWAS additionally adjusted for month of birth. To account for genetic relatedness and geographical structure in UKB, we applied a linear mixed model using the BOLT-LMM software to perform GWAS [41]. Genetic instruments from GWASs were selected based on variants that met the criteria of $P < 5 \times 10^{-8}$ and $r^2 < 0.001$ using a reference panel of $n$ = 10,000 randomly selected unrelated European participants from UKB [42].

**Genetic instruments for other measures of lifecourse body composition.**    Although the childhood body size measure in UKB aims to capture a surrogate measure of adiposity at age 10 (i.e., whether an individual was "thinner," "plumper," or "about average"), we sought to assess this by accounting for other measures of body composition at different stages in the lifecourse. Specifically, we sought to investigate whether childhood height, birth weight or fat-free mass index (FFMI) may be responsible for findings using the childhood body size instruments rather than adiposity at age 10. The same protocol described above was therefore repeated to identify genetic instruments in the UKB study for childhood height (field #1697), adult height (field #50), birth weight (field #20022), and FFMI (field #23101 divided by field #50 squared). Childhood and adult height measures were categorized in the same manner as their body size

counterparts, whereas birth weight was kept as a continuous trait to maximize sample sizes in analyses. All GWASs were adjusted for age, sex, and genotyping chip, with the exception of childhood height, which was additionally adjusted for month of birth.

**Genetic effect estimates on MRI-assessed measures of cardiac structure and function.** Genome-wide genetic variant effects on measures of cardiac structure and function were obtained from a previous GWAS of cardiac MRI-derived left ventricular measurements in 36,041 UKB participants who attended follow-up clinics [30]. These measures were LVEDV, LVESV, SV and LVEF. GWASs were undertaken using BOLT-LMM with adjustment for age, sex, year of birth, and MRI scanner's unique identifiers. Estimates from these GWASs were unadjusted for BMI and height, which is why they were selected over others available.

**Genetic effect estimates on MRI-assessed measures of abdominal organs.** We additionally obtained genome-wide estimates on 5 measures of abdominal organ traits [43]. These were liver volume, liver fat percentage, pancreas volume, pancreas fat percentage, and kidney volume. As a further analysis, we also extracted estimated on SAT and VAT volume. These GWASs were conducted using BOLT-LMM with adjustment for age, $age^2$, sex, imagine center, scan date, scan time, and genotyping batch.

**Genome-wide association studies of cardiomyopathy endpoints.** We obtained genome-wide results from a previously conducted GWAS of 1,816 cases of nonischemic cardiomyopathy and 388,326 controls from the UKB study. Details of this GWAS have been described previously [44]. In brief, cases were defined as patients with reported hospitalization or death due to dilated cardiomyopathy or left ventricular failure (defined as ICD10 codes I420, I421, I422, I501, or ICD9 code 4281) and an absence of CAD (defined based on ICD9 and ICD10 codes reported in **S1 Table**). Additionally, we applied our BOLT-LMM GWAS pipeline described above to derived genetic estimates on dilated and hypertrophic cardiomyopathy separately (based on ICD10 codes I420 and I421/I422, respectively) with adjustment for age and sex. An overview of all the GWAS datasets analyzed in this study can be found in **S1 Table**. Characteristics of these datasets can be found in **S2 Table**.

**Early life measures of cardiac structure from the Avon Longitudinal Study of Parents and Children.** ALSPAC is a population-based cohort investigating genetic and environmental factors that affect the health and development of children. The study methods are described in detail elsewhere [45,46]. In brief, 14,541 pregnant women residents in the former region of Avon, UK, with an expected delivery date between April 1, 1991 and December 31, 1992, were eligible to take part in ALSPAC. Detailed phenotypic information, biological samples, and genetic data, which have been collected from the ALSPAC participants, are available through a searchable data dictionary (http://www.bris.ac.uk/alspac/researchers/our-data). Written informed consent was obtained for all study participants. Ethical approval for this study was obtained from the ALSPAC Ethics and Law Committee and the Local Research Ethics Committees.

We obtained data from ALSPAC participants enrolled in the Growth Related effects in ALSPAC on Cardiac Endpoints (GRACE) substudy [47]. At mean age = 17.8 years (range = 16.3 to 20 years), participants underwent an echocardiogram test that obtained measures of left ventricular structure and function. These measures were then analyzed using linear regression with weighted genetic risk scores for childhood and adult body size both individually and together in the same model with adjustment for age and sex.

## Statistical analysis

**Univariable mendelian randomization.** First, we conducted 2-sample univariable MR to investigate the total effect of genetically predicted childhood body size on each of the MRI-

derived outcomes in turn. This was estimated using the IVW method [48] for initial analyses followed by the weighted median [49] and MR–Egger [50] methods as sensitivity analyses. This was to evaluate the robustness of our IVW estimates to horizontal pleiotropy, which is the phenomenon whereby genetic variants exert their effects on exposure and outcome via 2 separate biological pathways [51]. All univariable analyses were repeated for adult body size as well as all other exposures investigated in this study. F-statistics were derived for each set of instruments to assess whether findings may be prone to weak instrument bias.

**Multivariable mendelian randomization.** We next investigated the direct and indirect effect of childhood body size on each of the MRI-derived outcomes using 2-sample multivariable MR [12,13]. This involved including adult body size in our model along with childhood body size to simultaneously estimate their genetically predicted effects on each outcome in turn. This analysis was then repeated using the genetic instruments for childhood and adult height, allowing us to investigate whether results for body size were likely due to adiposity rather than simply being larger in childhood.

Further sensitivity analyses were also conducted estimating the direct effect of childhood body size using multivariable MR while accounting for both FFMI and birth weight. As described previously, we accounted for birth weight in this study to investigate whether an individual's body size in very early life (for example before age 5 in the lifecourse) may be responsible for the results identified using our childhood body size genetic instruments [17]. The focus of this study was on childhood body size at age 10 in the lifecourse, and as such, investigating the relationship between birth weight and cardiac structure was outside its scope. Furthermore, an appropriate study design for this research question would require an assessment of the effect of parental genotypes, which we did not have access to in UKB [34,35].

All MR analyses were undertaken in R (version 3.5.1) using the "TwoSampleMR" package [52]. Forest plots in this paper were generated using the R package "ggplot2" [53].

## Supporting information

**S1 Fig.** Forest plots illustrating **(A)** univariable and **(B)** multivariable MR effect estimates of childhood body size BMI and FFMI on measures of cardiac structure and function. The data underlying this figure can be found in S8 and S9 Tables. BMI, body mass index; FFMI, fat-free mass index; LV, left ventricular; MR, mendelian randomization; MRI, magnetic resonance imaging.
(PNG)

**S2 Fig.** Forest plots illustrating **(A)** univariable and **(B)** multivariable MR effect estimates of childhood body size BMI and BW on measures of cardiac structure and function. The data underlying this figure can be found in S10 and S11 Tables. BMI, body mass index; BW, birth weight; LV, left ventricular; MR, mendelian randomization; MRI, magnetic resonance imaging.
(PNG)

**S3 Fig.** Forest plots illustrating **(A)** univariable and **(B)** multivariable MR effect estimates of childhood and adult height on measures of cardiac structure and function. The data underlying this figure can be found in S12 and S13 Tables. LV, left ventricular; MR, mendelian randomization; MRI, magnetic resonance imaging.
(PNG)

**S1 Table. Overview of the datasets analyzed in this study.**
(XLSX)

**S2 Table. Characteristics for the exposures and outcomes analyzed in this study.**
(XLSX)

**S3 Table. Univariable MR analyses for childhood and adult body size on heart structure measures. MR, mendelian randomization.**
(XLSX)

**S4 Table. Univariable and multivariable MR analyses for childhood and adult body size on heart structure measures.** MR, mendelian randomization.
(XLSX)

**S5 Table. Validation univariable and multivariable MR analyses for childhood and adult body size in the ALSPAC cohort on heart structure measures.** ALSPAC, Avon Longitudinal Study of Parents and Children; MR, mendelian randomization.
(XLSX)

**S6 Table. Univariable MR analyses for childhood and adult body size on abdominal measures.** MR, mendelian randomization.
(XLSX)

**S7 Table. Univariable and multivariable MR analyses for childhood and adult body size on abdominal measures.** MR, mendelian randomization.
(XLSX)

**S8 Table. Univariable MR analyses for childhood body size and FFMI on heart structure measures.** FFMI, fat-free mass index; MR, mendelian randomization.
(XLSX)

**S9 Table. Univariable and multivariable MR analyses for childhood body size and FFMI on heart structure measures.** FFMI, fat-free mass index; MR, mendelian randomization.
(XLSX)

**S10 Table. Univariable MR analyses for childhood body size and BW on heart structure measures.** BW, birth weight; MR, mendelian randomization.
(XLSX)

**S11 Table. Univariable and multivariable MR analyses for childhood and BW on heart structure measures.** BW, birth weight; MR, mendelian randomization.
(XLSX)

**S12 Table. Univariable MR analyses for childhood and adult height on heart structure measures.** MR, mendelian randomization.
(XLSX)

**S13 Table. Univariable and multivariable MR analyses for childhood and adult height on heart structure measures.** MR, mendelian randomization.
(XLSX)

**S14 Table. Univariable and multivariable MR analyses for childhood body size and childhood height on heart structure measures.** MR, mendelian randomization.
(XLSX)

**S15 Table. Univariable MR analyses for childhood and adult body size on risk of cardiomyopathy outcomes.** MR, mendelian randomization.
(XLSX)

**S16 Table. Univariable and multivariable MR analyses for childhood and adult body size on risk of cardiomyopathy outcomes.** MR, mendelian randomization.
(XLSX)

## Acknowledgments

We are extremely grateful to all the families who took part in this study, the midwives for their help in recruiting them, and the whole ALSPAC team, which includes interviewers, computer and laboratory technicians, clerical workers, research scientists, volunteers, managers, receptionists, and nurses. The UK Medical Research Council and Wellcome (Grant ref: 217065/Z/19/Z) and the University of Bristol provide core support for ALSPAC. Genetic data were generated by Sample Logistics and Genotyping Facilities at the Wellcome Trust Sanger Institute and LabCorp (Laboratory Corporation of America) using support from 23andMe.

## Disclaimers

This research was conducted at the NIHR Biomedical Research Centre at the University Hospitals Bristol NHS Foundation Trust and the University of Bristol. The views expressed in this publication are those of the author(s) and not necessarily those of the NHS, the National Institute for Health Research or the Department of Health. This publication is the work of the authors and TGR will serve as guarantor for the contents of this paper.

## Ethics statement

Written informed consent was obtained for all study participants. Ethical approval for this study was obtained from the ALSPAC Ethics and Law Committee and the Local Research Ethics Committees, and data were accessed under application C2823. Data from the UKB were accessed under application #15825, which has obtained ethics approval from the Research Ethics Committee (REC; approval number: 11/NW/0382).

## Author Contributions

**Conceptualization:** Tom G. Richardson.

**Data curation:** Katie O'Nunain.

**Formal analysis:** Katie O'Nunain.

**Funding acquisition:** George Davey Smith.

**Methodology:** George Davey Smith, Tom G. Richardson.

**Project administration:** Tom G. Richardson.

**Resources:** Katie O'Nunain.

**Software:** Katie O'Nunain.

**Supervision:** Tom G. Richardson.

**Visualization:** Katie O'Nunain.

**Writing – original draft:** Katie O'Nunain, Tom G. Richardson.

**Writing – review & editing:** Katie O'Nunain, Chloe Park, Helena Urquijo, Genevieve M. Leyden, Alun D. Hughes, George Davey Smith, Tom G. Richardson.

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
