## [Editor Report · Decision Letter 0]

12 Apr 2022

Dear Tom, 

Thank you for submitting your manuscript entitled "Childhood body size has a long-term influence on later life heart structure: A lifecourse Mendelian randomization study" for consideration as a Research Article by PLOS Biology. I have been handling your manuscript since my colleague Lucas is currently out of the office on parental leave. 

Please accept my sincere apologies for the delay in getting back to you as we confirmed the previous Circulation reports and evaluated your submission with an academic editor with the relevant expertise. I'm writing to let you know that we would like to send a minor revision based on your responses to the previous reviews at Circulation.

However, before we can this, we need you to complete your submission by providing the metadata that is required for full assessment. To this end, please login to Editorial Manager where you will find the paper in the 'Submissions Needing Revisions' folder on your homepage. Please click 'Revise Submission' from the Action Links and complete all additional questions in the submission questionnaire.

Once your full submission is complete, your paper will undergo a series of checks in preparation for further consideration. Once your manuscript has passed the checks we'll be able to register the actual decision itself. To provide the metadata for your submission, please Login to Editorial Manager (https://www.editorialmanager.com/pbiology) within two working days, i.e. by Feb 11 2022 11:59PM.

Kind regards,

Richard

Richard Hodge, PhD

Associate Editor, PLOS Biology

rhodge@plos.org

On behalf of:

Lucas Smith

Associate Editor, PLOS Biology

lsmith@plos.org

PLOS

---

## [Editor Report · Decision Letter 1]

19 Apr 2022

Dear Tom,

Thank you for submitting your manuscript "Childhood body size has a long-term influence on later life heart structure: A lifecourse Mendelian randomization study" for consideration as a Research Article by PLOS Biology. Please accept my apologies for the delay in getting this decision to you due to the recent Easter holiday. As with all papers reviewed by the journal, yours was evaluated by the PLOS Biology editors as well as by an Academic Editor. We asked the Academic Editor whether they would be able to use the previous reviews to reach a decision on your manuscript without recourse to further review.

Based on these discussions with the Academic Editor, we will probably accept this manuscript for publication, provided you satisfactorily address the remaining data-related requests that I have provided below (points A-F):

(A) After discussions within the editorial team, we would like to consider your manuscript as a Discovery Report. The editorial criteria for this article type can be found at the following link. (https://journals.plos.org/plosbiology/s/what-we-publish#loc-discovery-report). During resubmission, we ask that you please change your submission to this article type. 

(B) We would to suggest the following modification to the title, to make it more compelling for our broad readership:

"A lifecourse Mendelian randomization study reveals that childhood body size has a long-term influence on later life heart structure"

(C) In your ethics statement, please include the approval number provided by the ethics committee that approved the study.

(D) You may be aware of the PLOS Data Policy, which requires that all data be made available without restriction: http://journals.plos.org/plosbiology/s/data-availability. For more information, please also see this editorial: http://dx.doi.org/10.1371/journal.pbio.1001797

- Supplementary files (e.g., excel). Please ensure that all data files are uploaded as 'Supporting Information' and are invariably referred to (in the manuscript, figure legends, and the Description field when uploading your files) using the following format verbatim: S1 Data, S2 Data, etc. Multiple panels of a single or even several figures can be included as multiple sheets in one excel file that is saved using exactly the following convention: S1_Data.xlsx (using an underscore).

- Deposition in a publicly available repository. Please also provide the accession code or a reviewer link so that we may view your data before publication.

Regardless of the method selected, please ensure that you provide the individual numerical values that underlie the summary data displayed in the following figures, as they are essential for readers to assess your analysis and to reproduce it.

Figure 2A-B

(E) Please also ensure that each of the relevant figure legends in your manuscript include information on *WHERE THE UNDERLYING DATA CAN BE FOUND*, and ensure your supplemental data file/s has a legend.

(F) Please ensure that your Data Statement in the submission system accurately describes where your data can be found and is in final format, as it will be published as written there. This includes referencing where the underlying data can be found in the Supplementary Information, as well as providing the accession numbers for any data deposited in public databases.

------------------

*Published Peer Review History*

*Early Version*

Kind regards,

Richard

Richard Hodge, PhD

Associate Editor, PLOS Biology

rhodge@plos.org

On behalf of:

Lucas Smith, PhD

Associate Editor, PLOS Biology

lsmith@plos.org

PLOS

---

## [Editor Report · Decision Letter 2]

3 May 2022

Dear Tom,

On behalf of my colleagues and the Academic Editor, Jason Locasale, I am happy to say that we can in principle accept your Discovery Report entitled "A lifecourse Mendelian randomization study highlights the long-term influence of childhood body size on later life heart structure" for publication in PLOS Biology, provided you address any remaining formatting and reporting issues. These will be detailed in an email that will follow this letter and that you will usually receive within 2-3 business days, during which time no action is required from you. Please note that we will not be able to formally accept your manuscript and schedule it for publication until you have completed any requested changes.

During this time, we also ask that you please update your Data Availability Statement in the submission system to the following statement, just for some additional clarity:

“Summary statistics from genome-wide association studies can be found as referenced in Supplementary Table 1 and additional underlying data supporting this paper can be found within the Supplementary Tables 2-16. All individual level data analysed in this study can be accessed via an approved application to ALSPAC (http://www.bristol.ac.uk/alspac/researchers/access/) and the UK Biobank study (https://www.ukbiobank.ac.uk/enable-your-research/apply-for-access)."

PRESS

Sincerely, 

Richard

Richard Hodge, PhD

Associate Editor, PLOS Biology

rhodge@plos.org

PLOS
